# A single-molecule counting approach for convenient and ultrasensitive measurement of restriction digest efficiencies

Yi Zhang[1]*, Takuro Nunoura[2], Daisuke Nishiura[3], Miho Hirai[1], Shigeru Shimamura[1], Kanako Kurosawa[1], Chieko Ishiwata[3], Shigeru Deguchi[2]

1 SUGAR Program, X-star, Japan Agency for Marine-Earth Science and Technology (JAMSTEC), Yokosuka, Japan, 2 Research Center for Bioscience and Nanoscience, Research Institute for Marine Resources Utilization, Japan Agency for Marine-Earth Science and Technology (JAMSTEC), Yokosuka, Japan, 3 Center for Mathematical Science and Advanced Technology, Japan Agency for Marine-Earth Science and Technology (JAMSTEC), Yokosuka, Japan

* zhangyi@jamstec.go.jp

**Data Availability Statement:** All relevant data are within the manuscript and its Supporting Information files.

## Abstract

Restriction endonucleases play a central role in the microbial immune system against viruses and are widely used in DNA specific cleavage, which is called restriction digestion, for genetic engineering. Herein, we applied digital cell-free protein synthesis as an easy-to-use orthogonal readout means to assess the restriction digest efficiency, a new application of digital bioassays. The digital counting principle enabled an unprecedentedly sensitive trace analysis of undigested DNA at the single-molecule level in a PCR-free manner. Our approach can quantify the template DNA of much lower concentrations that cannot be detected by ensemble-based methods such as gold-standard DNA electrophoresis techniques. The sensitive and quantitative measurements revealed a considerable variation in the digest efficiency among restriction endonucleases, from less than 70% to more than 99%. Intriguingly, none of them showed truly complete digestion within reasonably long periods of reaction time. The same rationale was extended to a multiplexed assay and applicable to any DNA-degrading or genome-editing enzymes. The enzyme kinetic parameters and the flanking sequence-dependent digest efficiency can also be interrogated with the proposed digital counting method. The absolute number of residual intact DNA molecules per microliter was concluded to be at least $10^7$, drawing attention to the residual issue of genetic materials associated with the interpretation of nucleases' behaviors and functions in daily genetic engineering experiments.

## Introduction

Restriction endonuclease (RE) was discovered as a microbial immune barrier against bacteriophage infection more than fifty years ago [1]. The sophisticated restriction system in bacteria and archaea specifically cuts the invading DNA that is not protected by the host methylation modification. A relatively recent discovery of the clustered regularly interspaced short

**Funding:** This work was supported by the Japan Society for the Promotion of Science (JSPS) in the form of a grant [KAKENHI JP18K14260] awarded to YZ. Microfabrication was conducted at Takeda Sentanchi Supercleanroom, The University of Tokyo, supported by "Nanotechnology Platform Program" of the Ministry of Education, Culture, Sports, Science and Technology (MEXT), Japan, (Grant Number JPMXP09F20UT0006) awarded to YZ and KK. The funders had no role in study design, data collection and analysis, decision to publish, or preparation of the manuscript.

palindromic repeats (CRISPRs) and CRISPR-associated (Cas) proteins as an acquired immunity diversified the defense arsenal through recognizing and cutting the viral DNA sequence that has been already incorporated into the host genome during the past infection [2], in analogy with the vaccination of higher organisms. Since the discoveries, various restriction enzymes recognizing specific DNA sequences have been extensively identified and are used as an indispensable research tool for molecular biology. An old research tool, gel electrophoresis, has long been applied in the laboratory to date for decades to assess the performance of this kind of DNA-cleaving tools. In brief, undigested DNA can be fluorescently visualized on the gel as a band by intercalating dyes [3]. The easy preparation and the fluorescent readout made it more convenient and safer than even older methods such as sucrose-gradient sedimentation and radioautography [4, 5], but the sensitivity of gel electrophoresis is becoming insufficient toward demanding applications such as ever-evolving DNA or RNA sequencing. It would be ideal if every single uncleaved DNA molecule can be distinguished from the dense background of cleaved ones.

Among existing commercially available platforms, digital PCR (dPCR) (a single DNA molecule can be amplified) and optical mapping (a single DNA molecule can be stretched and fluorescently imaged) [6] might be most likely capable of meeting this demand. However, to the best of our knowledge, nobody tried or intended to adopt dPCR for checking and optimizing their restriction digestion experiments. In the context of a dense background of identical DNA sequences, dPCR for a given restriction digest generally requires sophisticated design and expensive optimization for primers, probes, and thermal cycling conditions [7]. The optical mapping technology based on electrophoretic nanochannels, in general, can only detect DNA fragments larger than about two kilobases. The size resolution would be suitable for detecting genome-scale structural variations but is insufficient for cut-and-paste DNA manipulation in general genetic engineering experiments. An optically limited 100 bp resolution may be achieved but requires expensive super-resolution techniques [8]. As exemplified in our investigation later, restriction digest analysis sometimes requires size resolution even better than 100 bp. Hence, it is impractical to involve dPCR or sophisticated single-molecule imaging platforms in daily routine restriction digestion, simply due to the high financial and time costs, or the insufficient DNA sizing capability.

Cell-free synthetic biology is an emerging technological platform for biosensor development in recent years due to the low-cost, one-pot configuration, and ease-to-execute [9–12]. Our recent work enabled a massively parallel cell-free protein synthesis (CFPS) from single DNA molecules encapsulated in individual femtoliter droplets and has proved that every encapsulated DNA molecule can reliably trigger the CFPS reaction in the corresponding droplet [13]. If the template DNA is pre-treated by nucleases, cleavage on the DNA strand can result in premature transcripts and truncated polypeptides. In the present study, we will show that every single intact (i.e., undigested) DNA molecule in the dense background of identical DNA sequences can stably produce the CFPS signal without being impeded by the concomitant DNA digests. In this study, we intend to divert the droplet array CFPS system into detecting single DNA molecules, featuring advantages such as PCR-free, label-free (DNA without fluorescent labeling), heating-free (CFPS can be carried out at room temperature), sequence-specific (only the specified DNA can produce the corresponding functional protein), highly sensitive and quantitative (based on digital counting principle) [14, 15], and low-cost. Unlike dPCR, there is no need to use or optimize primers, probes, and thermal cycling; therefore, this PCR-free approach composed of only mixing, partitioning, and incubating at room temperature can be readily implemented in the lab and easily be extended to multiplexing, without costly optimization. The signal detection is based on conventional widefield microscopy. All of the features suggest that our approach would be superior to previous DNA fragment analysis

ones [16, 17]. This study applies several modular DNA sequences containing the restriction sites of interest and uses CFPS as a readout means to interrogate the restriction digest efficiency of a wide variety of REs and other DNA-degrading enzymes.

## Materials and methods

### Reagents

REs (NcoI, BtgI, XbaI, NdeI, NheI, BmtI, BamHI, XcmI, PflMI, BstEII, NcoI, HpaI, BbsI, BsgI, AfeI, BstXI, StuI, BsrGI, and EcoRI, accompanied with respective buffers) were from New England Biolabs (NEB). Another NdeI (accompanied with H buffer), and recombinant DNase I (RNase-free) were from Takara-Bio. Another NcoI (accompanied with H buffer) and DNase I were from Nippon Gene. Hi-Lo DNA marker was from Bionexus. DNA primers were synthesized by Thermo Fisher Scientific. Other chemicals were from FUJIFILM Wako Pure Chemical Corporation unless otherwise noted.

### Template DNA preparation

The gene of mNeonGreen [18], mTurquoise2 [19], and mScarlet [20] was synthesized by Eurofins Genomics and inserted into the pRSET-B vector using In-Fusion HD cloning kit (Takara-Bio), respectively. After transformation using *E. coli* JM109 competent cells (RBC Bioscience), the recombinant plasmid was extracted and purified using QIAprep spin miniprep kit (Qiagen). The plasmid sequences were validated by Sanger sequencing (Applied Biosystems 3130xl Genetic Analyzer or Applied Biosystems 3730xl DNA Analyzer, BigDye Terminator V3.1, and BigDye XTerminator Purification Kit, FASMAC). A forward primer (`GCGAAATTAATACGA CTCACTATAGGG`) and a reverse primer (`ACCCCTCAAGACCCGTTTAG`) were used to amplify the region containing the T7 promoter (T7P), ribosome binding site (RBS), and the marker gene. The PCR amplicon was purified using QIAquick PCR purification kit (Qiagen) and quantified using NanoDrop (Thermo Fisher Scientific). This linear DNA was used as the template of restriction digestion and CFPS reactions.

The upstream and downstream three nucleobases flanking BmtI site of mNeonGreen template were modified from original ATG (i.e., `ATGGCTAGCATG`) to TTT (i.e., `TTTGCTAGC TTT`) or GGG (i.e., `GGGGCTAGCGGG`), where the underlined region is BmtI recognition site, using QuikChange lightning site-directed mutagenesis kit (Agilent). The primer set for the TTT mutagenesis was `TTGCTGTCCACCAGTAAAGCTAGCAAAACCATGATGATGATGATG ATGAGAACC` and `GGTTCTCATCATCATCATCATCATGGTTTTGCTAGCTTTACTGGTGG ACAGCAA`, and the primer set for the GGG mutagenesis was `CCCATTTGCTGTCCACCAGTC CCGCTAGCCCCACCATGATGATGATGATGATG` and `CATCATCATCATCATCATGGTGGGGC TAGCGGGACTGGTGGACAGCAAATGGG`, where the underlined bases are the mutations. The sequence of both constructs was validated by Sanger sequencing (FASMAC). The prepared plasmids were also used as the PCR template to prepare the corresponding linear template DNA. The linear DNA was used as the template of restriction digestion and CFPS reactions.

### Restriction digestion by type II REs

The linear template DNA was digested following the manufacture's protocol. Unless noted otherwise, 1 μg template DNA, 5 μl 10× buffer, 10 U RE, and nuclease-free water (Thermo Fisher Scientific) were mixed to a final volume of 50 μl. The mixture was incubated at 37 °C on a thermal cycler (Veriti, Thermo Fisher Scientific), and 1 μl was pipetted from the mixture during the incubation at the time point of 1 min, 3 min, 5 min, 7 min, 15 min, 1 h, and 16 h,

respectively, as needed. The pipetted 1 μl solution was immediately diluted 100~4000 times in nuclease-free water.

## DNA cleavage by CRISPR-Cas9

Guide-it sgRNA in vitro transcription and screening system kit (Takara-Bio) was used to digest the linear mNeonGreen template DNA. For a comparison purpose, two forward primers (CCTCTAATACGACTCACTATAGGCGATGACGATAAGGATCCGAGTTTAAGAGCTATGC; CCTCTAATACGACTCACTATAGGTCGCCTTTCCAGGCCGCCAGTTTAAGAGCTATGC) were designed to prepare the DNA template used for the in vitro transcription of the corresponding guide RNA (sgRNA) targeting the respective sites (underlined sequences of the above primers) of mNeonGreen DNA template. The designed forward primer was mixed with the accompanied Guide-it scaffold template (containing the reverse primer) to amplify the sgRNA template using the accompanied PrimeSTAR Max DNA polymerase premix. The PCR amplicon was directly used as the template of the in vitro transcription of sgRNA using the accompanied Guide-it T7 polymerase mix and the corresponding buffer. The accompanied DNase I was added to the transcript solution to remove the template DNA. The transcribed sgRNA was purified using the accompanied Guide-it ITV RNA clean-up kit. 50 ng sgRNA was mixed and incubated with 250 ng Guide-it recombinant Cas9 nuclease at 37 $^{\circ}$C for 5 min. Then, 150 ng linear mNeonGreen DNA template, Cas9 reaction buffer, and bovine serum albumin were mixed to the Cas9/sgRNA solution in a final volume of 15 μl, and incubated at 37 $^{\circ}$C for 1 h, followed by an inactivation at 80 $^{\circ}$C for 5 min.

## DNA digestion by strandase

Strandase included in Guide-it long ssDNA production system kit (Takara-Bio) was used to digest single-stranded DNA (ssDNA). The linear mNeonGreen DNA was newly prepared using PCR (PrimeSTAR Max DNA polymerase) with either a 5'-phosphorylated reverse primer and the non-phosphorylated forward primer set, or a 5'-phosphorylated forward primer and the non-phosphorylated reverse primer set. The phosphorylated primers were purified by HPLC (Thermo Fisher Scientific). The sequence of primers was the same as the one used in the preparation of linear template DNA. The double-stranded DNA (dsDNA) amplicon was purified using the QIAquick PCR purification kit. 5 μg dsDNA was first digested using strandase A with the accompanied buffer A in a total volume of 50 μl at 37 $^{\circ}$C for 5 min, followed by an inactivation at 80 $^{\circ}$C for 5 min, and then digested using strandase B with the accompanied buffer B in a final volume of 101 μl at 37 $^{\circ}$C for 5 min, followed by another inactivation at 80 $^{\circ}$C for 5 min.

## DNA digestion by DNase I

DNase I (RNase-free) from Takara-Bio or Nippon Gene was used to digest the linear mNeonGreen template DNA at 37 $^{\circ}$C for 30 min, followed by an inactivation at 80 $^{\circ}$C for 2 min.

## DNA electrophoresis

The DNA digests were analyzed using Agilent 2100 Bioanalyzer. 1 μl of 33-fold diluted (with nuclease-free water) DNA digest was injected into each well of the High Sensitivity DNA chip. Based on the known concentration of the DNA ladder (Agilent), the quantification of every DNA sample in each lane was automatically carried out by the Bioanalyzer software. In parallel, the digested sample was checked using 1.2% slab agarose (Bio-Rad) gel electrophoresis with RedSafe (iNtRON) staining. The gel was imaged using a 365 nm (for dsDNA) or 254 nm (only for ssDNA) UV transilluminator and a CCD camera (BioDoc-It Imaging System, UVP).

## Microfabrication

A microchamber array was fabricated on a cover glass via the previously established microfabrication process [21]. The resulting diameter and depth of each microchamber were 4 μm and 3 μm, respectively. The device was assembled via sandwiching a piece of pre-cut (using STIKA SV-8, Roland) double-coated adhesive Kapton tape (Teraoka Seisakusho) with a customized slab glass (Rs-JAPAN) and the microfabricated microchamber array substrate. Two through-holes on the slab glass and the removed region of the tape formed a microchannel (3 mm × 19 mm × 85 μm), covering over $10^5$ microchambers.

## Digital cell-free protein synthesis

Fluorescent proteins were synthesized from single DNA molecules using PURExpress in vitro protein synthesis kit (NEB) in the femtoliter droplet array (FemDA). 6 μl solution A, 4.5 μl solution B, 0.3 μl murine RNase inhibitor (NEB), 2.7 μl nuclease-free water, and 1.5 μl pre-diluted (i.e., 100-fold in nuclease-free water) DNA sample were mixed (total 15 μl), resulting in a total dilution factor of $10^3$ for the DNA digest. For a multiplexed assay, each DNA sample was 1 μl, and the water volume was accordingly adjusted to maintain the total volume unchanged. This mixture solution was introduced into the microchannel. A sequential injection of AE-3000 (spiked with 0.1 wt% Surflon S-386, AGC) and Fomblin Y25/6 oil (Solvay) isolated and sealed microchambers, forming the femtoliter droplets. The CFPS reaction was carried out at room temperature.

## Imaging and data analysis

FemDA was scanned using an inverted epifluorescence microscope (Nikon Ti-E) with a filter set of 480/30 nm excitation, 505 nm dichroic, and 535/45 nm emission for mNeonGreen, a filter set of 430/25 nm excitation, 455 nm dichroic, and 470/25 nm emission for mTurquoise2, and a filter set of 560/40 nm, 595 nm dichroic, and 630/60 nm emission for mScarlet. A 60× or 100× objective lens (CFI Apo TIRF, NA = 1.49, Nikon) and a CCD camera (Andor iXon3 DU-897) was used for imaging. The exposure time was 100 ms. An automatic image analysis was carried out as previously reported [21]. Histogram plot and Gaussian fitting of the fluorescence intensities were done by KaleidaGraph (Synergy). Poisson fitting based on the method of least squares was carried out using an online tool (http://vassarstats.net/poissonfit.html). The time variation of template DNA concentration given by $S(t) = K_m W\{(S_0/K_m)\cdot\exp[(-V_{max}t+S_0)/K_m]\}$ was numerically solved by applying Newton's iterative method, where $W$ is the omega function (also called Lambert $W$ function), $S$ is the substrate concentration, $S_0$ is the initial concentration of substrate, $K_m$ is the Michaelis constant, $V_{max}$ is the maximum reaction rate, and $t$ is the reaction time. The numerical calculation was carried out on an Excel macro written by Visual Basic for Applications (VBA) language. The curve fitting with the Michaelis-Menten equation was done by GraphPad Prism. Other statistical analyses and drawings were done by Excel (Microsoft Office).

## Results

### General scheme for evaluating restriction digest efficiency at the single-molecule level

The overall study supposed that there are a small number of intact DNA molecules in the digest solution which cannot be detected by electrophoresis and proposed a new scheme to detect such undigested DNA molecules, as shown in Fig 1. The restriction site of interest is located within (or upstream) of the coding sequence of a marker gene fused into a modular

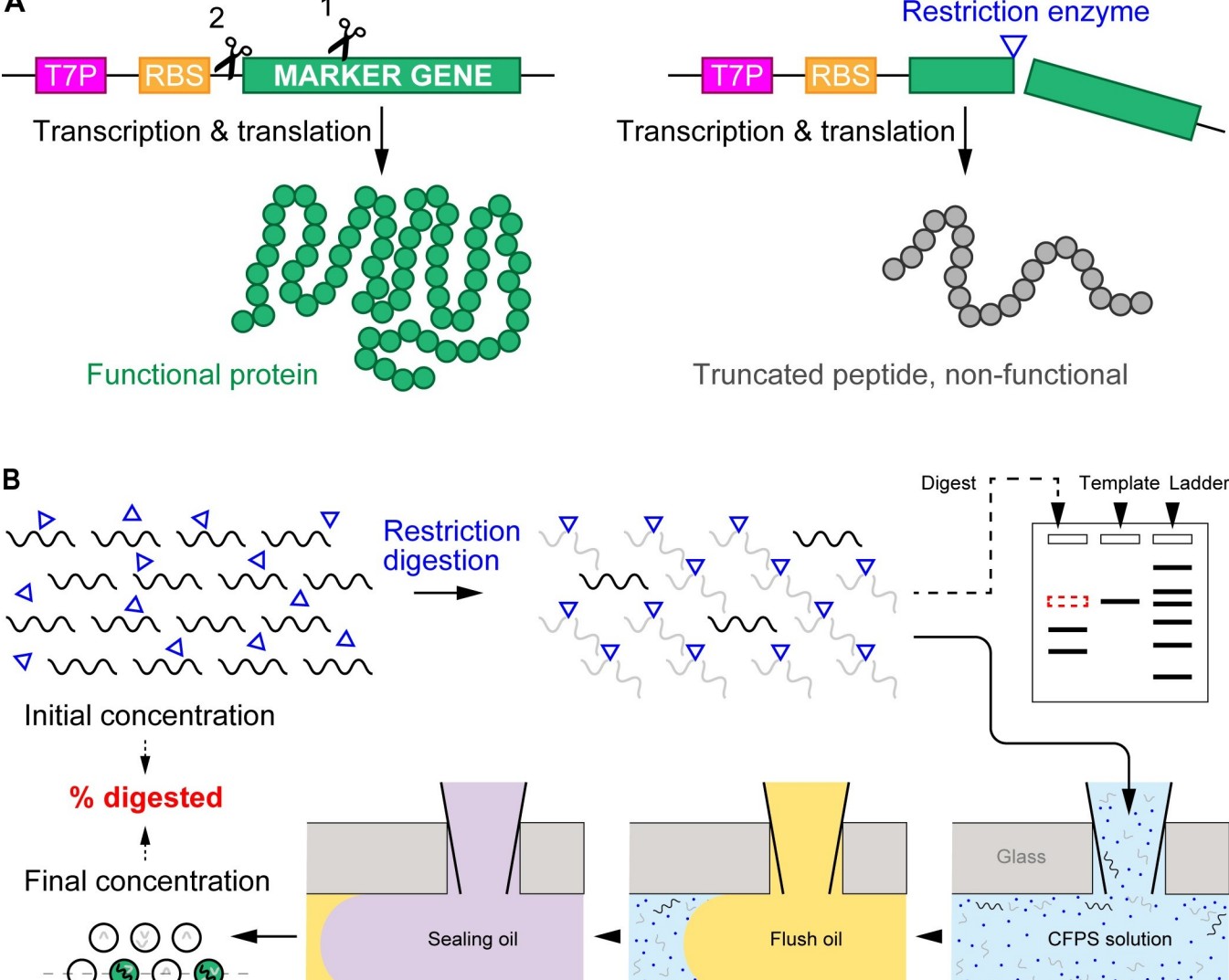

**Fig 1. A general strategy to measure restriction digest efficiency.** (A) DNA cleavage-triggered loss of function of marker proteins. A modular template DNA used for CFPS consists of T7P, RBS, and a marker gene encoding for fluorescent proteins. The restriction site of interest can be either within the marker gene (scissor 1) or upstream of the marker gene (scissor 2). The restriction digest terminates the peptide synthesis prematurely, hence no or only truncated peptides can be produced. (B) Basic experimental procedure for quantifying the undigested DNA molecules. A defined volume of the digest solution is directly mixed with CFPS aliquot. The mixture is introduced into FemDA. Only intact (i.e., undigested) DNA in the droplets can produce functional proteins that can be detected using fluorescence microscopy. The fraction of positive droplets over the array is used to derive the concentration of the undigested DNA. The residual template DNA that cannot be detected by gel electrophoresis (upper-right scheme) can be detected at the single-molecule level using FemDA. The dashed line on FemDA (lower-left) indicates a position of the cross-section view.

CFPS DNA template. The functional protein encoded by the marker gene cannot be synthesized without the integrity of the template DNA (Fig 1A). Furthermore, the massively parallel measurement in the droplet array followed by digital counting enables a quantitative assessment for the digest efficiency (Fig 1B). The experimental operation is only composed of mixing (the DNA digest solution with the aliquot of CFPS components), injecting (the mixture into

FemDA), incubating (the FemDA device), and imaging (with a standard microscope), which is simple enough to allow an inexpert to handle this system. A simple subtraction of the final DNA concentration from the initial concentration represents the extent of DNA digestion definitely.

This design provides users great flexibility on arbitrarily interrogating the REs of interest. In general, a marker gene (and its codon degeneracy) intrinsically contains many restriction sites (S1 Fig); therefore, it is feasible to use the same template to interrogate any one of them. Alternatively, the marker gene could be kept constant while only changing the restriction site of interest positioned upstream of the start codon of the marker gene (e.g., BamHI in the S1B Fig). The former way eliminates the effort for preparing the expression vector every time, and the later way eliminates the effort for searching a suitable marker gene containing the restriction site of interest.

The marker gene can be either a fluorescent protein or an enzyme [13]. We preferred to use fluorescent proteins because of the simplicity of the fluorescent protein that allows for a convenient batch processing of a set of restriction digests. The fluorescence intensity of fluorescent proteins faithfully reflects the number of template DNA molecules in the droplet, which is less dependent on reaction time. Thus, the batch CFPS can be carried out overnight, and only end-point single-frame imaging is required (S1 Movie). The previous studies have already proved that every single purified template DNA molecule in FemDA can trigger the CFPS reaction [13, 21] since the input concentration (lower than one molecule per droplet on average) of template DNA was consistent with the output ratio of fluorescent droplets (denoted as $P_{positive}$, which is equal to the number of positive droplets divided by the total number of droplets) over the array. In this study, the situation is slightly different from the previous ones as the intact single DNA molecules always coexist with the interferential DNA digest fragments even after partitioning the CFPS solution into droplets. The higher the digest efficiency, the lower the fraction of intact DNA in the droplet, and the higher the demand for the assay sensitivity. To confirm the robustness of the proposed biosensor, we first applied a blend of intact template DNA and gel-purified DNA digests (composed of equal molar concentrations of two DNA fragments) for a series of digital CFPS reactions. Specifically, the concentration of the intact DNA in the mix was kept constant (0.16 DNA molecules per droplet), while the DNA digest was serially diluted, resulting in a molar ratio of full-length DNA versus DNA digest from 3:924 to 3:1. Although the absolute fluorescence intensity of positive droplets was reduced with the increase of the amount of digested DNA, which is attributable to the inevitably partial consumption of resources on the truncated peptide synthesis, the $P_{positive}$ value remained constant (S2 Fig). $P_{positive}$ (0.15) was consistent with the intact DNA input concentration (i.e., $1-e^{-0.16} = 0.15$) even if the abundance of intact DNA was down to approximately 0.3% (i.e., equivalent to a restriction digest efficiency of 99.7%) (S2 Fig). Therefore, the high reaction efficiency in the femtoliter droplet system guarantees that every intact DNA molecule can produce sufficient functional protein molecules for epifluorescence detection even in the dense background of interferential DNA.

## Reaction time dependency of restriction digest efficiency

Our experiments detected the residual template DNA that may still exist in the digest solution over time. The initial experiment arbitrarily chose the NcoI restriction site, which is located near the middle of the mNeonGreen sequence. Fluorescent proteins are highly susceptible to their structural integrity. The truncated protein completely lost its ability to fluoresce (S3 Fig). Following a standard recipe, 1 μg template DNA was digested using NcoI-HF in CutSmart buffer at 37 $^o$C. 1 μl of the reaction mixture was transferred into 99 μl water (i.e., 100-fold

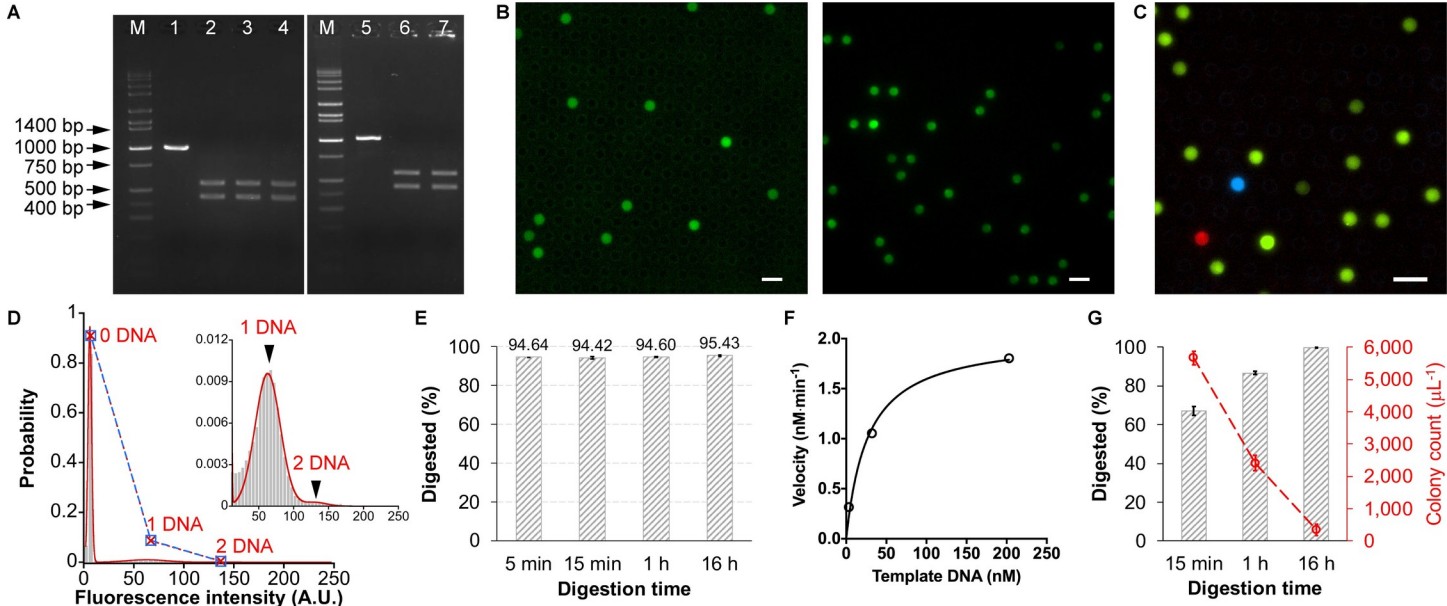

**Fig 2. Assessment for restriction digest efficiencies.** (A) Agarose gel electrophoresis of restriction digests. 80 ng DNA was applied to each lane. Lane M: DNA marker; lane 1&5: mNeonGreen template DNA (prior to digestion); lane 2: DNA digested by NcoI for 1 h; lane 3: DNA digested by NcoI-HF for 1 h; lane 4: DNA digested by NcoI-HF for 15 min; lane 6: DNA digested by NcoI-HF for 24 h; lane 7: DNA digested by BtgI for 24 h. (B) Digital CFPS using DNA digest solution. The NcoI-HF (left) or NdeI (right)-digested (for 15 min) mNeonGreen DNA solution was mixed with CFPS components and introduced into FemDA. Fluorescent proteins were synthesized in the droplet containing undigested DNA. Scale bar: 10 μm. (C) Simultaneous measurement of restriction digest efficiency of multiple enzymes with multiplexed digital CFPS. BamHI-HF-digested mTurquoise2 (cyan) DNA, NdeI-digested mNeonGreen (green) DNA, and EcoRI-HF-digested mScarlet (red) DNA were mixed and used for a single CFPS reaction on FemDA. The digestion time for each was 15 min. Scale bar: 10 μm. (D) Sum of Gaussian distributions of equal peak-to-peak intervals of fluorescence intensities of droplets. The NdeI-digested (for 1 h) DNA solution was used for this digital CFPS. The blue square represents the observed probability of droplets containing different numbers of template DNA molecules. The red cross represents the corresponding probability fitted by the Poisson distribution. The observed probability and the fitted probability were consistent with each other, proving the Poisson distribution of single DNA molecules. (E) Restriction digest efficiencies (digested versus the initial quantity of template DNA, expressed in %) of NcoI-HF at different time points. (E) Measurement of enzyme kinetics and Michaelis-Menten curve fitting for NdeI. (G) Restriction digest efficiencies (left y-axis marked in black) of NdeI at different time points and the number of false-positive colonies (right y-axis marked in red) caused by the undigested DNA in a model transformation experiment. Each reaction or transformation was performed in triplicate. Error bars: 1 standard deviation.

dilution) at the time point of 5 min, 15 min, 1 h, and 16 h, respectively. The digestion reaction in the diluted solution was virtually stopped due to the largely altered buffer and salt environment that are no longer suitable for the RE functioning. This is much simpler than a common heat-inactivation treatment (80 °C, 20 min), and no notable differences were found (S4 Fig). The heating-free treatment makes it advantageous over dPCR, in which exposure of DNA molecules to heating is thought to impair the amplification efficiency and hence result in false-negative [7]. Impressively, although the gel electrophoresis showed plausibly complete digestion (Fig 2A), the digital CFPS showed a certain fraction of positive droplets over the array (Fig 2B and S1 Movie). The sum of Gaussian distributions of equal peak-to-peak intervals of the fluorescence intensity of droplets over the array reinforced that the protein synthesis was triggered by single DNA molecules (Fig 2D) [13]. The frequency of droplets containing different numbers of DNA molecules was a perfect fit ($r^2 = 0.9999$) to a Poisson distribution, as expected for a random distribution of single DNA molecules. The digestion time was gradually extended, which is a widely-used countermeasure to improve the digest efficiency; however, few improvements were observed, even extending the time to 16 hours (Fig 2E). Interestingly, the 5-min digestion showed an almost identical digest efficiency as those of longer digestion times, which is consistent with the Time-Saver property of this enzyme qualified by its supplier, NEB. Another supplier (Nippon Gene)'s NcoI also showed a similar digest efficiency

(93.55% at 5 min, 93.68% at 15 min, 94.37% at 1 h, and 96.15% at 16 h), and the increments of the percentage over time were also slight. BtgI (NEB) recognizing the same restriction site did not show a significant improvement (96.29% digested) either after an extremely long time, 24 hours. Therefore, extending time may not be always as effective as we think of, at least for some REs.

Based on the digital counting principle [14], the exact concentration or the number of template DNA molecules remaining undigested in the digest solution can be calculated. In the above case of NcoI-based digestion, approximately 5% (1.79 nM) undigested DNA remained in the solution, corresponding to approximately $10^9$ copies per microliter (S1 Appendix). Although it is invisible in gel electrophoresis, this is a large amount of DNA in terms of the number of molecules.

The residual issue and its degree can also be theoretically and quantitatively predicted by steady-state enzyme kinetics. As demonstrated over the past decades, REs generally obey the Michaelis-Menten kinetics [22–29]. In the context of restriction digestion, the concentration of residual template DNA associated with the digestion time and initial DNA concentration can be expressed with an omega function (S5A Fig), an extended integral of the Michaelis-Menten equation along the reaction time [30]. The experimental data were in agreement with the theoretical values obtained from the omega function plugged with the respective reported Michaelis constants ($K_m$) and maximum reaction rates ($V_{max}$) (S5B–S5F Fig). This meta-analysis revealed a residual DNA concentration of 3 pM ~ 3 nM for a given DNA sample with an initial concentration of about 3 ~ 30 nM. This estimation was in good agreement with the digital counting results (see Table 1 later).

## Simplified measurement of enzyme kinetics based on digital counting

The proposed digital counting assay is primarily designed for endpoint measurement, quantifying the residual template DNA at specified time points. Although being a noncontinuous assay, it is possible to measure enzyme kinetics by combining the data from different time points. As summarized in S1 Table, the working concentration of most REs is much lower than the template DNA concentration (20 ng/μl = 32 nM), suggesting a multiple recycling of the enzyme in the steady-state rather than a single-turnover regime. We arbitrarily chose NdeI (standard working concentration: 9 nM) as a model case to determine the turnover number ($k_{cat}$) and the Michaelis-Menten constant ($K_m$). In this series of measurements, the initial concentration of mNeonGreen template DNA ranged from 203 nM (i.e.,126 ng/μl) to 3.2 nM (i.e., 2 ng/μl). The digest efficiencies at the time point of 1, 3, 5, and 7 min were determined using the digital CFPS for each DNA concentration. In stark contrast to conventional bulk measurement, where low background fluorescence or absorbance generally occurs, the digital assay is insusceptible to the low background of the reaction solution, which means that a perfect zero signal at time zero is always available. This unique feature compensates for the inability to carry out continuous measurements since the initial velocity of the enzymatic reaction can be immediately determined just after acquiring the first time-point data (i.e., the zero point data and the first time-point data can generate a reliable linear fitting for calculating the slope as well as the reaction velocity). The initial velocity at each concentration was thus derived using the first-minute data and constituted the reaction velocity vs. substrate concentration plot (Fig 2F). The $k_{cat}$ and $K_m$ of NdeI under the standard condition (i.e., NEB CutSmart buffer at 37 °C) were then determined to be 0.22 min$^{-1}$ and 27 nM, respectively, by a curve fitting based on the Michaelis-Menten equation. The relatively low $k_{cat}$ and relatively high $K_m$ can explain the low digest efficiency of this enzyme that we investigate in later sections.

**Table 1. Restriction digest efficiency of different REs.**

| Enzyme | $P_{positive}$[c] | Undigested template (nM)[d] | Digest efficiency (%) by digital counting[e] | Residual template DNA (pg) detected by Bioanalyzer[f] | Digest efficiency (%) by Bioanalyzer[g] | Residual template DNA (ng) loaded to slab gel[h] |
|---|---|---|---|---|---|---|
| XbaI[a] | 0.00543 | 0.24 | 99.25 | N.A. | N.A. | 0.75 |
| NdeI[a] | 0.240 | 10.56 | 67.12 | 223.88 | 59.9 | 32.88 |
| NheI[a, b] | 0.0287 | 1.27 | 96.06 | 20.86 | 96.3 | 3.94 |
| BmtI[a, b] | 0.195 | 8.60 | 73.23 | 182.59 | 72.8 | 26.77 |
| BamHI[a, b] | 0.0107 | 0.47 | 98.53 | 10.05 | 98.1 | 1.47 |
| XcmI | 0.0142 | 0.63 | 98.05 | N.A. | N.A. | 1.95 |
| PflMI[a] | 0.0289 | 1.27 | 96.03 | N.A. | N.A. | 3.97 |
| BstEII[a, b] | 0.0166 | 0.73 | 97.72 | 6.22 | 99.2 | 2.28 |
| NcoI[a, b] | 0.0410 | 1.79 | 94.42 | | | 5.58 |
| HpaI | 0.0303 | 1.34 | 95.84 | 3.82 | 99.4 | 4.16 |
| BbsI[a, b] | 0.0922 | 4.06 | 87.35 | 53.15 | 91.1 | 12.65 |
| BsgI[a] | 0.0383 | 1.69 | 94.74 | 4.12 | 99.3 | 5.26 |
| AfeI | 0.0351 | 1.55 | 95.18 | N.A. | N.A. | 4.82 |
| BstXI[a] | 0.162 | 7.14 | 77.76 | 245.22 | 61.5 | 22.24 |
| StuI[a] | 0.0452 | 1.99 | 93.80 | 8.91 | 98.5 | 6.20 |
| BsrGI[a, b] | 0.0453 | 2.00 | 93.78 | 14.50 | 97.7 | 6.22 |

[a]Time-Saver version (qualified by the supplier NEB).

[b]High-fidelity version (qualified by the supplier NEB [31]) of the enzyme.

[c]Fraction of positive droplets over the array (the number of positive droplets divided by the total number of droplets).

[d]The final concentration was calculated based on the digital counting principle (i.e., $P_{positive}/(v \cdot N_A)$, where $v$ is the droplet volume and $N_A$ is the Avogadro constant) and the sample dilution factor (1000).

[e]Final concentration of template DNA divided by its initial concentration (20 ng/μl, or 32.12 nM).

[f]Values adopted from the report of Bioanalyzer. N.A. = not available.

[g]Peak concentration divided by the region concentration. The concentration values were from the report of Bioanalyzer.

[h]The final concentration of template DNA multiplied by the gel-loading sample volume (5 μl).

## Multiplexed assay for simultaneous measurement of multiple restriction digest efficiencies

The modular template DNA used for the digital CFPS is compatible with many kinds of fluorescent proteins. This property offered a possibility for easy multiplexing. As a proof-of-concept, a 3-plexed measurement was carried out by mixing three DNA digests selected arbitrarily, BamHI-HF-digested mTurquoise2 (cyan fluorescent protein) template DNA, NdeI-digested mNeonGreen (green fluorescent protein) template DNA, and EcoRI-HF-digested mScarlet (red fluorescent protein) template DNA, to a single CFPS reaction on FemDA (Fig 2C). BamHI-HF, NdeI, and EcoRI-HF showed 99.49%, 65.69%, and 99.53% digest efficiency, respectively. The dynamic range is good enough for most measurements of restriction digest efficiency since the demonstrated multiplexed assay may have already covered the highest and the lowest efficiencies in a large population of representative REs (see Table 1 and the following sections later). It would be worthy of remembering the fact that EcoRI-HF showed the highest digest efficiency in 15 min for a single cut among many tested REs during the period of this study. EcoRI is one of the earliest discovered REs and has been most extensively subjected to sophisticated crystallography and protein engineering works in the past decades; hence the high cleavage efficiency is supposed to be a reward for the relevant studies.

## Comprehensive analyses of restriction digest efficiencies

As mentioned above, the proposed scheme can conveniently compare tens of different REs using the same DNA template. The next experiment traversed the unique restriction sites of mNeonGreen template DNA using the corresponding RE one by one. Each restriction digestion was carried out with its optimal buffer at standard (template DNA 1 μg/50 ul = 32 nM) identical reaction conditions for 15 min. The residual template DNA was detected and analyzed using the singleplex digital CFPS. None of them showed complete digestion (see $P_{positive}$ in Table 1). Surprisingly, the proportion of the cleaved DNA molecules ranged from less than 70% to more than 99%, a considerable variation that may exceed a common estimate. In consequence, the proportion of the residual template DNA ranged from 0.75%~32.88%, corresponding to $1.5 \times 10^8 \sim 6.4 \times 10^9$ copies/μl. Another supplier (Takara-Bio)'s RE was used to repeat the digestion with NdeI that exhibited the lowest digest efficiency in the above experiments, and a similar result ($P_{positive}$ = 0.211, 71.04% digested, 28.96 ng residual template DNA) was obtained. When the marker protein was changed from mNeonGreen to *Cobetia marina* alkaline phosphatase, and its template DNA (1 μg/50 μl = 19 nM) was digested with the NdeI (NEB), a similar digest efficiency ($P_{positive}$ = 0.136, 68.71% digested) was also obtained. Thus, the different digest efficiencies are more likely from the respective endonucleases' intrinsic property than from the supplier.

As DNA electrophoresis is the current "gold-standard" technique for checking DNA digests worldwide, we compared the single-molecule counting approach with two representative gel electrophoresis techniques: slab gel electrophoresis and capillary electrophoresis (CE) (Fig 3). The former is popular in general laboratories, and the latter is more sensitive than the former but with high cost. The template DNA in some digests (NdeI, BmtI-HF, BbsI-HF, BstXI) were clearly visible as a single band on both CE chips (Figs 3 and S6) and slab gel (Fig 3) since the quantity of the template DNA in these digests was greater than the respective limit of detection (LOD) of each method. However, as shown in the electropherograms, most of the template DNA remaining undigested were difficult or unable to detect, due to the insufficient LOD of electrophoresis. Specifically, the residual template DNA in the groups of NheI-HF, BamHI-HF, and BsrGI-HF were faint on the CE chip, while the residual template in the groups of XcmI, PflmI, BstEII-HF, HpaI, BsgI, AfeI, and StuI were invisible or undetectable (Fig 3). The quantity (refer to Table 1) of template DNA in these barely detectable groups (i.e., NheI-HF, BamHI-HF, BstEII-HF, HpaI, BsgI, StuI, and BsrGI-HF) can be considered close to (slightly higher than, or slightly lower than) the LOD of CE. A minimum detectable quantity of 25.5 ± 11.2 pg can thus be derived using the digital counting results of such groups, instead of using the software-calibrated relative values, which is reasonably consistent with the sensitivity of CE systems (Bioanalyzer or Fragment Analyzer) [32, 33]. Similarly, the slab gel electrophoresis (Fig 3 and Table 1) suggested a minimum detectable quantity of 5.0 ± 0.9 ng DNA, which is also consistent with the LOD of agarose gel electrophoresis [34]. Thus, the digital counting method is not only seamlessly compatible with electrophoresis but also can reliably detect DNA of much lower concentrations that cannot be detected by electrophoresis (Table 1). The consistent results between FemDA and CE also indicated that every single undigested DNA in FemDA could be generally expressed to the protein; otherwise, the digital counting result would deviate from the CE counterpart. Capillary or slab gel electrophoresis is also powerless for DNA fragments with too similar sizes (e.g., the XbaI-digested DNA sample) that can only be analyzed by the function-based assay. It is noteworthy that only 38% (5/13) of the Time-Saver qualified REs showed over 95% (equivalent to 5 ng residual per 100 ng total DNA) digestion; unexpectedly, all (3/3) of the non-Time-Saver qualified ones (XcmI, HpaI, AfeI) showed the digestion beyond 95% (Table 1).

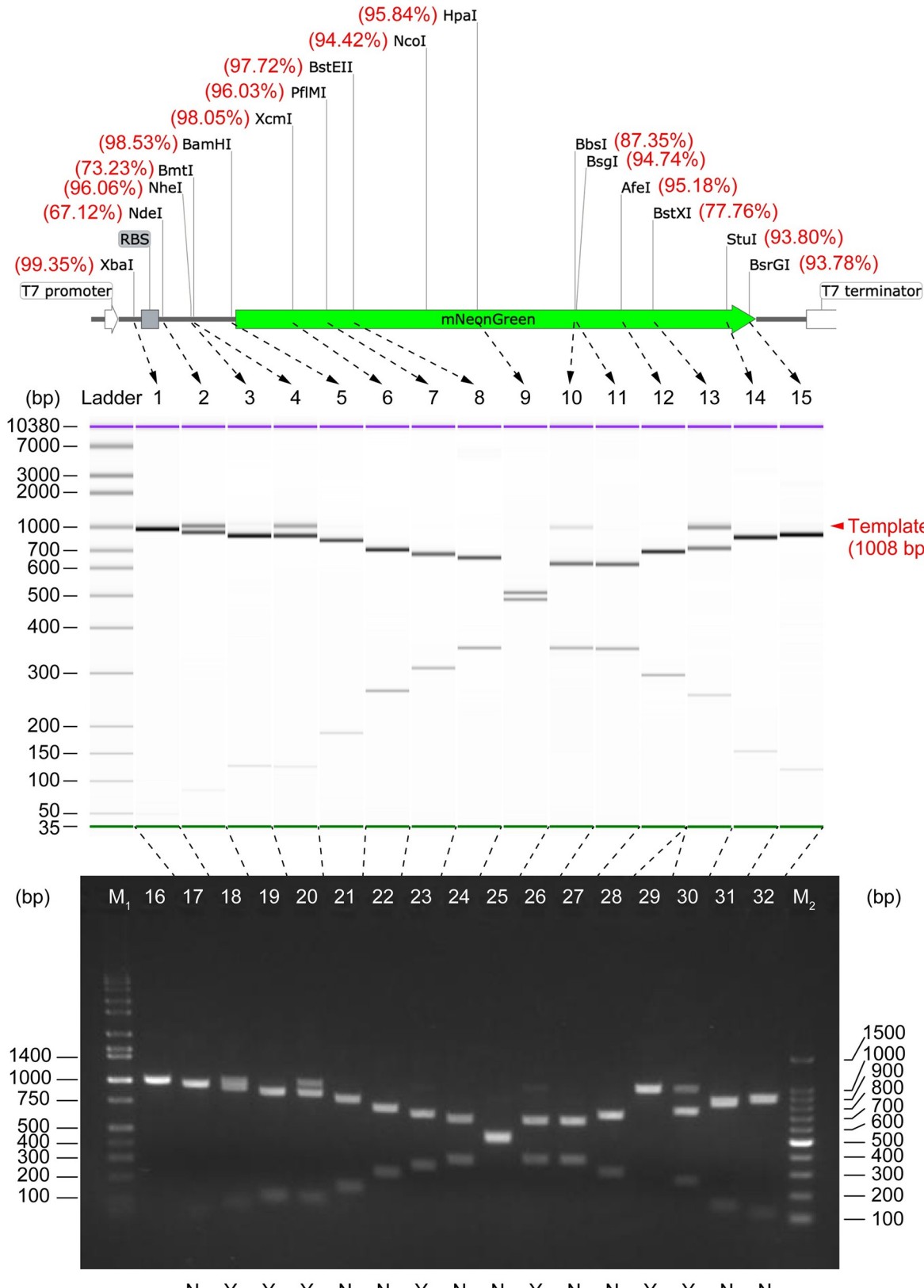

**Fig 3. DNA digest analysis using capillary electrophoresis (Agilent Bioanalyzer) and slab gel electrophoresis.** The measured digest efficiency (in % unit, from Table 1) was labeled beside the corresponding RE site on the template DNA map. On capillary electrophoresis (middle), lane 1~15: mNeonGreen DNA restriction digest with (in the order from left to right) XbaI, NdeI, NheI-HF, BmtI-HF, BamHI-HF, XcmI, PflMI, BstEII-HF, HpaI, BbsI-HF, BsgI, AfeI, BstXI, StuI, and BsrGI-HF, respectively. On agarose slab gel electrophoresis (bottom), lane $M_1$: Hi-Lo DNA marker; lane 16: template DNA (prior to digestion, 1008 bp); lane 2~17: digested by (in the order from left to right) XbaI, NdeI, NheI-HF, BmtI-HF, BamHI-HF, XcmI, PflMI, BstEII-HF, HpaI, BbsI-HF, BsgI, AfeI (newly purchased), AfeI (old stock), BstXI, StuI, BsrGI-HF; lane $M_2$: 100 bp DNA ladder; label "Y": undigested DNA was detectable on the slab gel; label "N": undigested DNA was not detectable on the slab gel. 100 ng (calculated based on the stock concentration of template DNA solution measured using NanoDrop) DNA was applied to each lane of the slab gel.

As the digestion using NdeI was too far from complete, the experiment tried to extend its digestion time, just as what we did for NcoI. The digest efficiency gradually increased over time (Fig 2G). Although the digestion reaction seemed slow, 99.71% template DNA was finally cleaved, which was the highest percentage among our tested ones. Even though, the approximately 92 pM final concentration of the residual template DNA still has a high probability of resulting in false positives in a cloning experiment (S2 Appendix). This possibility was confirmed by transforming the DNA digest into *E. coli* competent cells (Fig 2G). The colony number decreased with the increase in digest efficiency, but it was far from zero (S7 Fig). It should be noted that DNA transformation is only a qualitative measurement, which is susceptible to the transformation efficiency of the host cell, the plasmid size, and the incubation/culture conditions.

The sequence surrounding the target site is often supposed to influence cleavage efficiency. To experimentally investigate this possibility, we attempted to modulate the digest efficiency through modifying the GC-content of the sequence flanking the BmtI site, another site outside the mNeonGreen coding region. The corresponding RE also showed relatively poor digestion (73% digested, see Table 1) when the original GC-content of upstream and downstream three bases (see S1 Fig) was 33%. The digest efficiency was improved to 81% if the GC-content was changed to 0% (i.e., both upstream ATG and downstream ATG were all changed to TTT). Conversely, the digest efficiency can be lowered to 60% if the GC-content was changed to 100% (both ATGs were all changed to GGG). Hence, an inverse relationship between digest efficiency and GC-content was observed (at least for BmtI), which can be reasonably attributed to the ease of a local unwinding of DNA duplex for enzyme binding, moving, or cleavage.

Two long-term stored REs (HpaI manufactured in 2001, and AfeI manufactured in 2003) were used for the digestion test. The HpaI showed a digestion result comparable to others (Table 1), while the AfeI showed an inability to function (Lane 29 in Fig 3). Although both were properly stored in a freezer at -30 °C, they showed a completely different performance after nearly twenty years, suggesting different stabilities. By the way, the digital CFPS was still pursued with the failure digest of AfeI. Its $P_{positive}$ was 0.49, well consistent with the Poisson-predicted value assuming no digestion occurred in the initial 20 ng/µl DNA solution (S3 Appendix). This accidental instance well reflected the reliability and accuracy of the digital counting device.

It should be noted that the digestion with BsgI, a type IIs enzyme that has been suspected to require at least two copies of the recognition site on the DNA strand for its optimal cleavage activity [35], cleaved the template DNA very efficiently, which is even comparable to most of the single-site enzymes. The mechanistic and stereochemical details about such REs may be worthy of further in-depth studies.

## Digest efficiency of various kinds of DNA-degrading enzymes

The modular DNA template consists of the minimum genetic elements required for CFPS. Any DNA degradation, including nicks, breaks, and deletions, occurred on the sense or

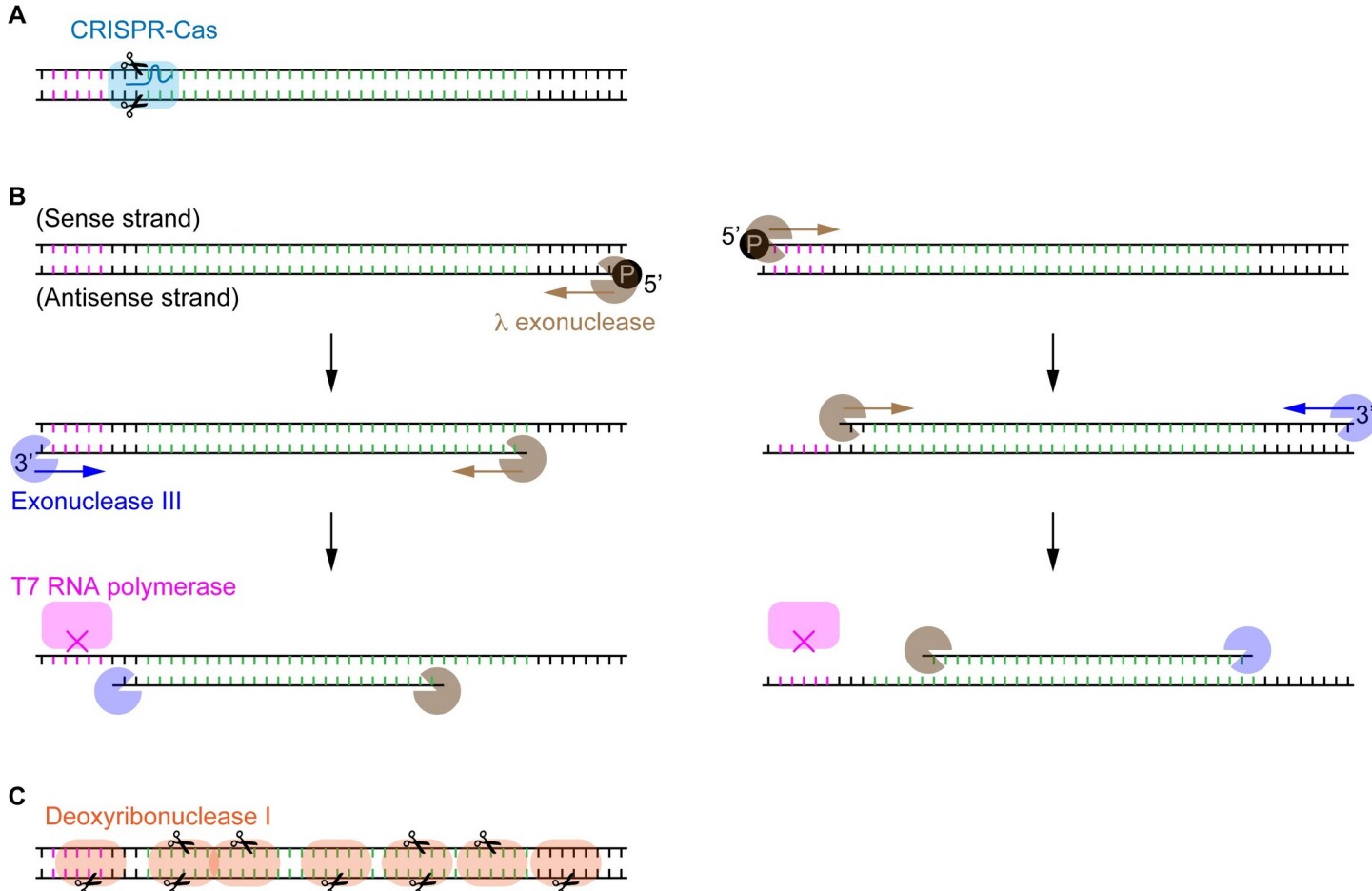

**Fig 4. The modular template DNA to interrogate the digest efficiency of various DNA-cleaving/degrading enzymes.** The protein-coding region is colored in green. (A) Site-specific cleavage of DNA by CRISPR-Cas nucleases with the aid of sgRNA. (B) ssDNA preparation using exonucleases. Both of λ exonuclease and exonuclease III recognize dsDNA's specific ends and digest one of the strands along the chain. This consecutive digestion disables the transcription and translation, as the duplex DNA sequence of the T7P region (pink colored) is essential for the binding of T7 RNA polymerase. (C) Non-specific cleavage of DNA by DNase I. Either the truncated open reading frame or the defective promoter or 5' untranslated region can lead to the premature termination of transcription and translation.

antisense strand generally disables the transcription or translation (Fig 4). Thus, the extent of degradation associated with the relevant enzymes can be quantified based on the digital assay scheme.

The experiment first interrogated the digest efficiency of CRISPR-Cas9 nuclease (also known as the type V restriction enzyme), a part of the prokaryotic immune system against virus infection. An efficient cleavage is the premise of a successful genome editing. Quantitative estimation of the cleavage efficiency is particularly of great importance to an emerging field of somatic cell genome editing [36], as this kind of non-reproductive cells remaining unmodified would retain the original functionalities and compromise the desired editing effect through cell division. The Cas9-sgRNA complex bound and cleaved DNA at the specific site (Fig 4A), and produced two fragments with different sizes that can be resolved by gel electrophoresis (S8A Fig). For a comparison purpose, two different guide RNA (sgRNA) targeting the respective sites of the mNeonGreen template were designed. One Cas9-sgRNA targeting the site around NcoI/BtgI cleaved the DNA inefficiently, and the other one targeting the site around BamHI seemed to cleave the DNA efficiently as the template DNA after 1 h cleavage

was invisible on the gel. The cleavage solution was similarly subjected to the digital CFPS on FemDA. 0.25 nM template DNA was uncleaved, which is equivalent to a cleavage efficiency of 98.44%. The CRISPR-Cas9 with an optimal sgRNA possesses the digest efficiency similar to some high-efficiency type II REs.

The next experiment analyzed the digest efficiency of strandase, a mixture of end group-specific exonucleases. Owing to the length limitation in chemical synthesis of oligo DNA, strandase as a cost-effective research tool is particularly useful for preparing long ssDNA. In addition to applications in microarrays, DNA sequencing, and biosensors, ssDNA gained considerable attention in genome editing in recent years as ssDNA has distinct advantages over the dsDNA counterpart in terms of low cytotoxicity and reduced off-target random integration [37]. In this sense, ssDNA's purity after the dsDNA digestion is vital for a reliable interpretation of experimental results. The proprietary product with the trade name of strandase is composed of two separate exonucleases. One behaves as λ exonuclease to selectively digest 5'-phosphorylated strand of the DNA duplex leaving the complementary ssDNA intact, and the other one behaves as exonuclease III to selectively digest one strand of the DNA duplex from 3'-OH end. If the digestion with λ exonuclease is carried out earlier, the later exonuclease III eventually only cleaves the same strand because exonuclease III is inactive on 3'-protruding end (Fig 4B). The consecutive digestion using λ exonuclease and exonuclease III disables the transcription and translation through either destroying the duplex structure of T7P or truncating the protein-coding sequence. The enzymatic digestion looked perfect on the gel electrophoresis (S8B Fig); however, over one hundred fluorescent droplets were detected on FemDA. This is equivalent to a digest efficiency higher than 99.99%.

As the dual digestion suggested an improved digest efficiency over the single-site digestion, we speculated that the more cleavage sites, the fewer residual template DNA. The next experiment used deoxyribonuclease I (DNase I) to digest the mNeonGreen template DNA (Fig 4C). DNase I can nonspecifically cleave the DNA strand on multiple sites. Except for some possible cleavage occurred at the very terminal bases, any one of cut actions can result in abortion of functional protein synthesis. Zero fluorescent droplets were found among over $5\times10^4$ droplets. This result was consistent across different sources of DNase I. The digestion with DNase I was the only case showing a perfect digest during this study while it was not restriction reactions.

## Discussion

The PCR-free and electrophoresis-free workflow is generally preferred in future laboratory technology development to eliminate the potential risk of DNA aerosol contamination [38]. Contrary to the relative quantification based on electrophoresis or real-time PCR, digital counting allows direct and absolute quantification for the DNA sample without DNA makers or calibration standards. FemDA featuring low sample consumption and superior LOD considerably lowered the required minimum quantity of DNA samples. It is worth trying to adopt some promising low-cost cell-free systems [12] and microdevice mass production to make the total cost close to the usual slab gel method.

Seeing the unseen is the charm and the aim of the evolving bioanalytical sensors. Any detection method must have its own LOD. The superior LOD primarily benefits from the zero background of the orthogonal digital assay scheme. The array consisting of $5\times10^4$ droplets has a theoretical LOD of 0.88 fM, equivalent to appropriately 500 DNA molecules (i.e., 0.5 fg for 1 kb dsDNA) per microliter. Although the converted mass concentration is variable and dependent on the molecular weight, the digital bioassay may have $10^3$ times greater sensitivity than capillary electrophoresis (5 pg) and $10^6$ times greater than agarose slab gel electrophoresis (5 ng), respectively. Researchers always pursue complete restriction digestion through optimizing

the reaction recipe and estimate the degree of completion of digestion reactions based on gel electrophoresis. The insufficient LOD of electrophoresis must always be carefully taken into account particularly for the judgment of a small amount of DNA, which is essential, sometimes highly crucial (e.g., forensic identification), but might be often overlooked [39]. For example, the claim of complete digestion on many brochures of commercially available REs are all based on gel electrophoresis and thus should not be misread as truly 100% completion.

The essence of RE is an enzyme, implying the enzymatic reaction can only reach an equilibrium state instead of the complete consumption of reactants. Our results provided direct evidence that extending the time for DNA digestion is often less effective than the expected when the enzymatic reaction plateaus quickly or the enzyme is less stable at high temperature. Extending time is effective only for some restriction digestions in which the corresponding enzyme must be stable enough over the extended time. It should always be taken into account that researchers generally face the false positives in the molecular cloning experiment. In general, no matter what the researcher does, there would always be over $10^7$ undigested template DNA molecules per microliter (assuming a 99.9% digest efficiency, much higher than the average in this study), a surprising number that may greatly exceed our expectations.

Apart from being a useful molecular biology tool, the single-molecule detection may offer a unique perspective to view the restriction system in vivo, i.e., the host-parasite interactions and their coevolution. Our results suggested a possibility of a ubiquitous residual viral DNA during the rounds of defense and offense. Hence, in addition to potential inhibition/escaping mechanisms based on anti-CRISPRs [40], the enzyme kinetics may be the most irresistible and naturally occurring forces against an ideal complete elimination of viruses. This point that has been neglected in the past could (at least partially) explain a seemingly contradictory phenomenon of the high cleavage efficiency and the ubiquitous presence of viruses in the microbial communities [41, 42]. As a consequence, archaeal and bacterial cells may generally carry viruses [43]. In particular, the stochastic nature of protein expression in individual cells during the period of cell division inevitably leads a portion of cells to contain only a small number of protein molecules [44]. One protein molecule or one viral DNA molecule in 1 fl volume (similar to a bacterial cell) has 1.67 nM concentration. Depending on the specific values of the kinetic constants (e.g., typical $K_m$ is approximately $10^0$ to $10^2$ nM, typical $k_{cat}$ is approximately 1 $min^{-1}$) of the particular RE [45], the viral DNA molecules have a chance to survive the digestion reaction within the bacterial doubling time (typically < 1 h) (S9A and S9B Fig). Nevertheless, the chance of escaping immune system may be slim since the concentration of RE molecules in a suitable host cell may be over $10^3 \sim 10^4$ nM [46, 47], large enough to rapidly (within a few minutes) cleave $10^3$ DNA molecules (S9C and S9D Fig), and the doubling time may be much longer in the wild than in the laboratory [48]. The trans-cleavage activity or the RNase activity of some stimulated CRISPR-Cas systems may also reinforce the integrated immune defense against foreign nucleic acids [49, 50]. The residual issue confirmed in vitro using our approach can provide precise stoichiometric insights into the virus-host interactions, where the number of virus particles, the concentration of restriction-modification enzymes, and the infection/treatment time must play their respective roles. This kind of knowledge may guide new antiviral therapy and prophylaxis, and call attention to the possibly large number of unedited target sites in the genome-editing experiments.

Our tool can deepen the basic understanding of the REs not only for the canonical type II REs that were discussed in detail in this study but also for other emerging types (e.g., base editors) and their inhibitors [51]. Several unexpected results found in this study reflected the insufficient knowledge regarding this category of enzymes in spite of the fact that they were discovered for over fifty years and were highly commercialized. For instance, two or more copies of the recognition sequence may not be essential for an efficient cleavage for some type IIs

enzymes (e.g., BsgI). Also, many restriction enzymes empirically require additional nucleobases flanking the target recognition site for efficient DNA cleavage, but the composition of such bases has few been defined. As revealed in the systematic site-directed mutagenesis experiments, decreasing the GC-content of just a few flanking bases may offer a simple way to improve the cleavage efficiency, particularly for efficient cleavage close to the very end of PCR products used for subcloning. The ultrasensitive and quantitative analytical technology may offer important clues for modifying or discovering mechanisms regarding interactions of DNA enzymes with their substrates.

## Conclusions

Digital protein synthesis in massively parallel droplets enabled single-molecule DNA detection in a high-throughput, quantitative, amplification-free, and label-free manner. This study demonstrated the power of cell-free biosensor in the ultrasensitive and quantitative measurement of the restriction digest efficiency for a variety of DNA-degrading enzymes. The direct evidence proved that truly complete digestion is generally impossible. For technical extensions, a reversed process based on signal-on (instead of the signal-off) scheme can be readily applied for interrogating the ligation efficiency of various ligases, making this tool valuable for broader genetic engineering studies. The single marker gene of the fluorescent protein or enzyme could be replaced by genetic circuits with specific output signals to increase the biosensor's functionality and extend the application range in the future.

## Supporting information

**S1 Appendix. A theoretical number of template DNA molecules remaining undigested.**
(PDF)

**S2 Appendix. A theoretical number of false positive clones in molecular cloning experiments.**
(PDF)

**S3 Appendix. $P_{positive}$ calculation.**
(PDF)

**S1 Fig. DNA templates and restriction sites.** (A) An overview of the expression plasmid maps for mNeonGreen, mScarlet, and mTurquoise2. The translucent blue-shaded region is amplified using PCR to generate the linear DNA used for restriction digestion and CFPS reactions. (B) The sequences of the linear template DNA for mNeonGreen, mScarlet, and mTurquoise2. Unique 6+ cutters are marked on the sequence.
(TIF)

**S2 Fig. Consistent $P_{positive}$ values independent of the concentration of digested DNA.** Full-length (1008 bp) mNeonGreen template DNA was prepared with PCR, purified from the PCR solution, and quantified by NanoDrop (133 ng/uL). It was diluted 1000 times in $H_2O$ to an arbitrary concentration of 214 pM, which is equivalent to 0.16 DNA molecules per droplet if adding 0.5 μL to a 15 μL CFPS mix (i.e., further diluted 30 times). In parallel, two fragments in a NcoI-digested DNA solution were separated by gel electrophoresis, purified from the gel for each, and quantified by NanoDrop. The relatively large fragment (~550 bp) and the relatively small fragment (~440 bp) had different mass concentrations but had similar molar concentrations ($6.6\times10^4$ pM). These two DNA fragments were serially diluted in $H_2O$. Each of the (diluted or undiluted) fragments was added 0.5 μL to the CFPS mix, resulting in a molar ratio of full-length DNA: each DNA fragment from 214: $6.6\times10^4$ to 214: $6.6\times10^1$, i.e., from 1: 308 to

3: 1. The experiment did not further increase the proportion of the digested DNA because the available concentration of gel-purified DNA is limited in practice. As shown in the box plot (blue colored), the absolute fluorescence intensity of individual positive droplets was reduced with the increase of the proportion of digested DNA, which is attributable to the resource consumption on the truncated peptide synthesis. The blue cross marker in the box is the mean of each group. In spite of the co-encapsulation of the interferential DNA (in this case, the small fragment containing T7P and RBS) in the droplet, the proportion of positive droplets remained unchanged (red colored) and was consistent with the input concentration of the full-length template DNA (theoretical $P_{positive} = 1 - e^{-0.16} \approx 0.15$). All molecular weights were pre-calculated using an online tool http://biotools.nubic.northwestern.edu/OligoCalc.html. (TIF)

**S3 Fig. Comparison of fluorescence intensity between intact and truncated mNeonGreen.** The mNeonGreen template was digested by NcoI-HF, and the N-terminus fragment was purified from the agarose gel. The intact and the truncated template DNA contain the same T7P, RBS, and the sequences located upstream of the start codon. An equal quantity (68 ng) of DNA was added into a 15 μL CFPS solution. The CFPS reaction was carried out in a microtiter plate (Non-binding, μClear, 384-well plate, Greiner Bio-One) at room temperature for a sufficiently long time (10 h). The fluorescence intensity was recorded using a microplate reader (Synergy, BioTek) with an excitation filter (485/20 nm) and an emission filter (528/20 nm). The run-off transcript from the NcoI-digested template was translated to a truncated polypeptide. In contrast to the intact mNeonGreen, the truncated one showed no fluorescence increase over time. (TIF)

**S4 Fig. Effect of heat-inactivation treatment.** (A) Digital protein synthesis using NcoI-digested mNeonGreen DNA solution without heat-inactivation. (B) Digital protein synthesis using NcoI-digested mNeonGreen DNA solution with heat-inactivation (80 °C, 20 min). Discrete distribution of fluorescence intensity can be observed in both cases, as a result of the stochastic distribution of DNA molecules into the droplets. From the histogram, the fraction of positive droplets (containing undigested DNA molecules) in both cases was the same with each other ($P_{positive\ [heat\text{-}inactivation]} = 3112/81408 \approx 0.038$; $P_{positive\ [heating\text{-}free\ inactivation]} = 1533/40693 \approx 0.038$). These results suggested there is no remarkable difference between heat-inactivation and heating-free inactivation. (TIF)

**S5 Fig. Time-dependent enzyme kinetics described with the aid of omega function.** (A) The concentration of residual template DNA (vertical axis) associated with the digestion time (horizontal axis) and initial DNA concentration (depth axis). We applied the enzyme kinetic constants ($K_m$ and $V_{max}$) that were determined by previous studies to the omega function of BamHI (B) [27], TaqI (C) [24], EcoRI (D, E) [26, 28], and PaeR7 (F) [25], respectively. The kinetic constants for S5A Fig were arbitrarily from the report of BamHI. Only a tiny portion of restriction endonucleases' kinetic constants have been measured in the past. The initial concentrations (as shown at the time-point zero of the horizontal axis) of template DNA were also taken from the respective reports. The determination of kinetic constants of restriction endonucleases is highly susceptible to the buffer components, reaction conditions, DNA sequences, and so on. As exemplified by the well-studied EcoRI (S5D Fig), our theoretical prediction was in good agreement with a previously reported real-time cleavage measurement [26], reflecting the reliability of the omega function. From this numerical calculation, we can speculate that there should be 3 pM ~ 3 nM DNA remaining undigested for a given DNA sample with an

initial concentration of about 3 ~ 30 nM. This estimation was in good agreement with our digital counting results (see Table 1).
(TIF)

**S6 Fig. Capillary electrophoresis for several representative restriction digests produced with unique cutters targeting mNeonGreen DNA.** (A) Capillary electrophoresis by Fragment Analyzer (Advanced Analytical Technologies). The same sample was analysed by Bioanalyzer in parallel (see Fig 3 in the main text). The reconstructed gel image clearly showed that the result of Fragment Analyzer was essentially identical to the result of Bioanalyzer. (B) Electropherograms of each DNA sample. Each group is composed of two electropherograms. The left one is Bioanalzyer's electropherogram, and the right one is Fragment Analyzer's electropherogram. The peak number, the relative peak height, and the relative peak intervals all indicated the identical performance between Bioanalyzer and Fragment Analyzer for analyzing the restriction digests.
(TIF)

**S7 Fig. Plasmid transformation using DNA digests.** The mNeonGreen expression plasmid was digested by NdeI for 15 min (left), 1 h (middle), and 16 h (right), respectively. An equal quantity of each digest was added into an equal aliquot of competent cells and plated together onto an ampicillin-selective LB-agar plate. The upper three are bright-field images, and the lower three are the corresponding fluorescence images under the illumination of 365 nm ultraviolet light.
(TIF)

**S8 Fig.** Agarose gel electrophoresis for DNA digests generated by CRISPR-Cas9 (A) or strandase (B). (A) Lane M: 100 bp DNA ladder; lane 1 & 3: linear mNeonGreen template DNA; lane 2: cleaved by Cas9-sgRNA targeting a site around BamHI, producing two DNA fragments about 180 bp and 820 bp; lane 4: cleaved by Cas9-sgRNA targeting another site around NcoI/BtgI, producing two DNA fragments about 440 bp and 560 bp. (B) Lane M: Hi-Lo DNA marker; lane 1: linear mNeonGreen template DNA phosphorylated at 5'-end of the antisense strand; lane 2: sense ssDNA; lane 3: linear mNeonGreen template DNA phosphorylated at 5'-end of the sense strand; lane 4: antisense ssDNA. In general, fluorogenic intercalators stains the ssDNA much less efficiently than for dsDNA. All lanes were loaded with 100 ng DNA sample.
(TIF)

**S9 Fig. Theoretical discussions about the restriction digest efficiency in a bacterial cell-sized compartment (1 fL).** Differing from the instances listed in S5 Fig, the calculation here was based on an "putative" enzyme, instead of the experimentally measured ones. The turnover number of the putative enzyme was assumed to be 1 min$^{-1}$. The viral DNA molecule was assumed to contain only one cleavage site. The amount of uncleaved DNA can be largely affected by the number of viral DNA molecules, the number of enzyme molecules, and the Michaelis constant ($K_m$). We gave herein several representative situations as "one DNA molecule and one enzyme molecule (A)", "many DNA molecule and a few enzyme molecules (B)", "many DNA molecules and many enzyme molecules (C)", and "much more DNA molecules and much more enzyme molecules (D)". Because the calculation for the case of $K_m$ = 1 nM exceeded the calculation capacity of Microsoft Excel, we only showed the other two in S9D Fig.
(TIF)

**S1 Movie. End-point imaging for CFPS of mNeonGreen on FemDA.** An NdeI-digested DNA solution was directly used as the source of template DNA. After an overnight incubation

for the CFPS, the droplet array was imaged using a fluorescence microscope.
(AVI)

**S1 Table. Working concentrations of our investigated restriction enzymes.**
(XLSX)

**S1 Raw images.**
(PDF)

## Acknowledgments

We thank Naoki Yoshida (New England Biolabs) for offering the enzyme concentration information.

## Author Contributions

**Conceptualization:** Yi Zhang.

**Data curation:** Daisuke Nishiura, Chieko Ishiwata.

**Formal analysis:** Yi Zhang, Takuro Nunoura, Daisuke Nishiura.

**Funding acquisition:** Yi Zhang.

**Investigation:** Yi Zhang, Miho Hirai, Shigeru Shimamura, Kanako Kurosawa.

**Methodology:** Yi Zhang.

**Project administration:** Yi Zhang.

**Resources:** Takuro Nunoura, Daisuke Nishiura, Kanako Kurosawa, Shigeru Deguchi.

**Software:** Daisuke Nishiura, Chieko Ishiwata.

**Supervision:** Takuro Nunoura, Shigeru Deguchi.

**Validation:** Yi Zhang, Miho Hirai.

**Visualization:** Yi Zhang, Daisuke Nishiura.

**Writing – original draft:** Yi Zhang.

**Writing – review & editing:** Yi Zhang, Takuro Nunoura, Kanako Kurosawa, Shigeru Deguchi.

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
