## [Decision Letter · Decision Letter 0]

28 Oct 2020

PONE-D-20-31271

A single-molecule counting approach for convenient and ultrasensitive measurement of restriction digest efficiencies

PLOS ONE

Dear Dr. Zhang,

Thank you for submitting your manuscript to PLOS ONE. After careful consideration, we feel that it has merit but does not fully meet PLOS ONE’s publication criteria as it currently stands. Therefore, we invite you to submit a revised version of the manuscript that addresses the points raised during the review process.

We look forward to receiving your revised manuscript.

Kind regards,

Ruslan Kalendar, PhD

Academic Editor

PLOS ONE

Journal Requirements:

2. We note that you have a patent relating to material pertinent to this article. Please provide an amended statement of Competing Interests to declare this patent (with details including name and number), along with any other relevant declarations relating to employment, consultancy, patents, products in development or modified products etc. Please confirm that this does not alter your adherence to all PLOS ONE policies on sharing data and materials, as detailed online in our guide for authors http://journals.plos.org/plosone/s/competing-interests by including the following statement: "This does not alter our adherence to  PLOS ONE policies on sharing data and materials.” If there are restrictions on sharing of data and/or materials, please state these. Please note that we cannot proceed with consideration of your article until this information has been declared.

Reviewers' comments:

Reviewer's Responses to Questions

**Comments to the Author**

1. Is the manuscript technically sound, and do the data support the conclusions?

Reviewer #1: Yes

Reviewer #2: Yes

2. Has the statistical analysis been performed appropriately and rigorously? 

Reviewer #1: Yes

Reviewer #2: Yes

3. Have the authors made all data underlying the findings in their manuscript fully available?

Reviewer #1: Yes

Reviewer #2: Yes

4. Is the manuscript presented in an intelligible fashion and written in standard English?

Reviewer #1: Yes

Reviewer #2: Yes

5. Review Comments to the Author

Reviewer #1: Zhang and colleagues present a fascinating technique using a femtoliter array for investigating DNA-cleaving enzymes. It would certainly be of interest if this microfabricated array could be mass produced as not all of us have access to clean-room manufacturing.

The actual technique for making and usinfg the array really needs a figure of its own to make the process clear.

They find that the restriction enzymes investigated do not give full cleavage as assessed by cell free protein synthesis. It is a pity that more detailed kinetic studies were not performed. It is essential that the enzyme concentrations are clearly stated - using "activity units" is not satisfactory as I am sure that some of these enzymes will have concentrations in excess of the DNA concentration in the digest thus the assays will be in a single-turnover regime while others may not be. (New England Biolabs do have this sort of information if asked) The lack of full cleavage by restriction enzymes is well known hence the development of the HF variants. It is also well known that the sequence surrounding the target site has an influence on cleavage efficiency - this should be investigated by analysing their template sequence.

The template sequences for the Scarlet and Turquoise genes need to be added to figure S1.

Figure S5 has a typographical error - it should have nM/min rather than nm/min.

Reviewer #2: Zhang et al., present a high sensitivity, facile and inexpensive method of assessing DNA digest or degradation efficiency. They use an elegant digital cell free protein synthesis assay as a read out of digest completeness for a variety of DNA restriction and DNA degrading enzymes and demonstrate for this system a low limit of detection. The system features a PCR free method without the need for heat inactivation or expensive instrumentation. The manuscript is well written, the figures well designed and the work well done. I have only a few minor amendments, listed below that should improve the clarity of the presentation.

General comments:

I would argue that the manuscript should feature one additional data figure in the main body of the text that compares the electrophoresis (currently shown in SFig 9) vs. the digital cell free protein synthesis results shown in tabular form. This would present data of interest to many general readers in a non-textual manner and without the need to access the supplementary information.

In the last paragraph of the discussion a single high efficiency digest of a type IIS RE is used to generalize that consensus that this class of RE requires two copies may be incorrect. This conclusion seems unjustified based on the results shown.

Minor edits:

-p21 line 326 Michalis-Menton should be Michaelis-Menton

-p24 line 376 ‘a single band on both of CE chips..’ should read‘a single band on both CE ’ chips…’

-p29 line 493 laboratorial should read as the simpler laboratory

-p30 line 524 ‘may greatly exceed our thought’ should read ‘may greatly exceed our expectations’

-p31 line 533 ‘As a consequence, archaea and bacteria prefer or have to coexist…’ the intended meaning of this sentence is very unclear.

6. PLOS authors have the option to publish the peer review history of their article (what does this mean?). If published, this will include your full peer review and any attached files.

Reviewer #1: No

Reviewer #2: No

---

## [Author Response · Author response to Decision Letter 0]

8 Dec 2020

Journal Requirements:

Response: We prepared the manuscript and named each file according to the provided style templates.

2. We note that you have a patent relating to material pertinent to this article. Please provide an amended statement of Competing Interests to declare this patent (with details including name and number), along with any other relevant declarations relating to employment, consultancy, patents, products in development or modified products etc. Please confirm that this does not alter your adherence to all PLOS ONE policies on sharing data and materials, as detailed online in our guide for authors http://journals.plos.org/plosone/s/competing-interests by including the following statement: "This does not alter our adherence to PLOS ONE policies on sharing data and materials.” If there are restrictions on sharing of data and/or materials, please state these. Please note that we cannot proceed with consideration of your article until this information has been declared.

Response: We added the patent name and application number and included the statement “This does not alter our adherence to PLOS ONE policies on sharing data and materials” in the amended statement of Competing Interests. Now, the statement is:

I have read the journal's policy and the authors of this manuscript have the following competing interests: [YZ and TN are inventors on a pending patent (Name: methods of obtaining the digestion efficiency of nucleases; Application number: 2019-232639) related to this work.] This does not alter our adherence to PLOS ONE policies on sharing data and materials.

Response: We provided the requested original images in a new supporting information file named as “S1_raw_images”.

Reviewers' comments:

Reviewer's Responses to Questions

Comments to the Author

1. Is the manuscript technically sound, and do the data support the conclusions?

Reviewer #1: Yes

Reviewer #2: Yes

2. Has the statistical analysis been performed appropriately and rigorously?

Reviewer #1: Yes

Reviewer #2: Yes

3. Have the authors made all data underlying the findings in their manuscript fully available?

Reviewer #1: Yes

Reviewer #2: Yes

4. Is the manuscript presented in an intelligible fashion and written in standard English?

Reviewer #1: Yes

Reviewer #2: Yes

5. Review Comments to the Author

Reviewer #1: Zhang and colleagues present a fascinating technique using a femtoliter array for investigating DNA-cleaving enzymes. It would certainly be of interest if this microfabricated array could be mass produced as not all of us have access to clean-room manufacturing.

Response: We thank the reviewer for the positive evaluation of our work and the interest in our array device. We are seeking funding to upgrade the microfabrication process and realize mass production.

The actual technique for making and usinfg the array really needs a figure of its own to make the process clear.

Response: We modified Fig. 1B to make the experiment process clear.

They find that the restriction enzymes investigated do not give full cleavage as assessed by cell free protein synthesis. It is a pity that more detailed kinetic studies were not performed. It is essential that the enzyme concentrations are clearly stated - using "activity units" is not satisfactory as I am sure that some of these enzymes will have concentrations in excess of the DNA concentration in the digest thus the assays will be in a single-turnover regime while others may not be. (New England Biolabs do have this sort of information if asked) The lack of full cleavage by restriction enzymes is well known hence the development of the HF variants. It is also well known that the sequence surrounding the target site has an influence on cleavage efficiency - this should be investigated by analysing their template sequence.

Response: This comment consists of two aspects. One is about the enzyme kinetics study, and the other one is about the investigation of sequence-dependent cleavage efficiency. 

For the first one, we got the concentration information (as shown in a new S1 Table in the Supporting Information) from NEB and showed the applicability of our method for the enzyme kinetics measurement. The result was summarized in a new sub-section entitled “Simplified measurement of enzyme kinetics based on digital counting”. 

For the second one, we modified the sequence surrounding the target site and confirmed the influence on cleavage efficiency using the digital counting method. The result was added to the sub-section “Comprehensive analyses of restriction digest efficiencies”.

Accordingly, we updated the Acknowledgments section and added some words to the Materials and Methods section for the above contents. Based on the newly gained insight, we also added some words to the Discussion section.

The template sequences for the Scarlet and Turquoise genes need to be added to figure S1.

Response: We added the template sequences of mScarlet and mTurquoise2 genes to Figure S1 and revised the figure caption accordingly.

Figure S5 has a typographical error - it should have nM/min rather than nm/min.

Response: This typographical error in Figure S5 was corrected.

Reviewer #2: Zhang et al., present a high sensitivity, facile and inexpensive method of assessing DNA digest or degradation efficiency. They use an elegant digital cell free protein synthesis assay as a read out of digest completeness for a variety of DNA restriction and DNA degrading enzymes and demonstrate for this system a low limit of detection. The system features a PCR free method without the need for heat inactivation or expensive instrumentation. The manuscript is well written, the figures well designed and the work well done. I have only a few minor amendments, listed below that should improve the clarity of the presentation.

Response: We appreciate the reviewer for the very positive evaluation of our work.

General comments:

I would argue that the manuscript should feature one additional data figure in the main body of the text that compares the electrophoresis (currently shown in SFig 9) vs. the digital cell free protein synthesis results shown in tabular form. This would present data of interest to many general readers in a non-textual manner and without the need to access the supplementary information.

Response: We offered a new Fig. 3 in the main text that combined the electrophoresis results and the digital counting results (i.e., the digest efficiency data). I guess that the “SFig 9” in the reviewer’s comment should be read as “S7 Fig”, since the tabular form does not contain the digital counting results of S9 Fig. The new Fig. 3 presented the data “to general readers in a non-textual manner and without the need to access the supplementary information”.

In the last paragraph of the discussion a single high efficiency digest of a type IIS RE is used to generalize that consensus that this class of RE requires two copies may be incorrect. This conclusion seems unjustified based on the results shown.

Response: We modified the sentence to make the argument more precise than before.

Minor edits:

-p21 line 326 Michalis-Menton should be Michaelis-Menton

Response: The typo was corrected.

-p24 line 376 ‘a single band on both of CE chips..’ should read‘a single band on both CE ’ chips…’

Response: The sentence was revised.

-p29 line 493 laboratorial should read as the simpler laboratory

Response: The word was revised.

-p30 line 524 ‘may greatly exceed our thought’ should read ‘may greatly exceed our expectations’

Response: The word was revised.

-p31 line 533 ‘As a consequence, archaea and bacteria prefer or have to coexist…’ the intended meaning of this sentence is very unclear.

Response: The sentence was improved to make the meaning clear.

A marked-up copy of our manuscript that highlighted changes made to the original version was upload as a separate file labeled “Revised Manuscript with Track Changes”. An unmarked version of our revised paper without tracked changes was upload as a separate file labeled “Manuscript”.

---

## [Editor Report · Decision Letter 1]

10 Dec 2020

A single-molecule counting approach for convenient and ultrasensitive measurement of restriction digest efficiencies

PONE-D-20-31271R1

Dear Dr. Zhang,

We’re pleased to inform you that your manuscript has been judged scientifically suitable for publication and will be formally accepted for publication once it meets all outstanding technical requirements.

Kind regards,

Ruslan Kalendar, PhD

Academic Editor

PLOS ONE

---

## [Editor Report · Acceptance letter]

22 Dec 2020

PONE-D-20-31271R1 

A single-molecule counting approach for convenient and ultrasensitive measurement of restriction digest efficiencies 

Dear Dr. Zhang:

I'm pleased to inform you that your manuscript has been deemed suitable for publication in PLOS ONE. Congratulations! Your manuscript is now with our production department. 

Kind regards, 

on behalf of

Prof. Ruslan Kalendar 

Academic Editor

PLOS ONE